# PRUNE: Preserving Proximity and Global Ranking for Network Embedding

**Yi-An Lai** [*‡]
National Taiwan University
b99202031@ntu.edu.tw

**Chin-Chi Hsu** [†‡]
Academia Sinica
chinchi@iis.sinica.edu.tw

**Wen-Hao Chen** [*]
National Taiwan University
b02902023@ntu.edu.tw

**Mi-Yen Yeh** [†]
Academia Sinica
miyen@iis.sinica.edu.tw

**Shou-De Lin** [*]
National Taiwan University
sdlin@csie.ntu.edu.tw

## Abstract

We investigate an unsupervised generative approach for network embedding. A multi-task Siamese neural network structure is formulated to connect embedding vectors and our objective to preserve the global node ranking and local proximity of nodes. We provide deeper analysis to connect the proposed proximity objective to link prediction and community detection in the network. We show our model can satisfy the following design properties: scalability, asymmetry, unity and simplicity. Experiment results not only verify the above design properties but also demonstrate the superior performance in learning-to-rank, classification, regression, and link prediction tasks.

## 1  Introduction

Network embedding aims at constructing a low-dimensional latent feature matrix from a sparse high-dimensional adjacency matrix in an unsupervised manner [1–3, 6, 15, 18–21, 23, 24, 26, 31].

Most previous works [1–3, 6, 15, 18–20, 23, 31] try to preserve $k$-order proximity while performing embedding. That is, given a pair of nodes $(i, j)$, the similarity between their embedding vectors shall be to certain extent reflect their $k$-hop distances (e.g. the number of $k$-hop distinct paths from node $i$ to $j$, or the probability that node $j$ is visited via a random walk from $i$). Proximity reflects local network topology, and could even preserve global network topology like communities. There are some other works directly formulate node embedding to fit the community distributions by maximizing the modularity [21, 24].

Although through experiments some of the proximity-based embedding methods had visualized the community separation in two-dimensional vector space [2, 3, 6, 18, 20, 23], and some demonstrate an effective usage scenario in link prediction [6, 15, 19, 23], so far we have not yet seen a theoretical analysis to connect these three concepts. The first goal of this paper is to propose a proximity model that connects node embedding with link prediction and community detection. There has been some research focusing on a similar direction. [24] tries to propose an embedding model preserving both proximity and community. However, the objective functions for proximity and community are separately designed, not showing the connection between them. [26] models an embedding approach considering link prediction, but not connect it to the preservation of the network proximity.

---

[*]Department of Computer Science and Information Engineering

[†]Institute of Information Science

[‡]These authors contributed equally to this paper.

Besides connecting link prediction and proximity, here we also argue that it is beneficial for an embedding model to preserve a network property not specifically addressed in the existing research: global node importance ranking. For decades unsupervised node ranking algorithms such as PageRank [16] and HITS [10] have shown the effectiveness in estimating global node ranks. Besides ranking websites for better search outcomes, node rankings can be useful in other applications. For example, the Webspam Challenge competition [4] requires that spam web pages to be ranked lower than non-spam ones; the WSDM 2016 Challenge [5] asks for ranking papers information without supervision data in a billion-sized citation network. Our experiments demonstrate that being able to preserve the global ranking in node embedding can not only boost the performance of a learning-to-ranking task, but also a classification and regression task training from node embedding as features.

In this paper, we propose Proximity and Ranking-preserving Unsupervised Network Embedding (PRUNE), an *unsupervised* Siamese neural network structure to learn node embeddings from not only community-aware proximity but also global node ranking (see Figure 1). To achieve the above goals, we rely on a generative solution. That is, taking the embedding vectors of the adjacent nodes of a link as the training input, the shared hidden layers of our model non-linearly map node embeddings to optimize a carefully designed objective function. During training, the objective function, for global node ranking and community-aware proximity, propagate gradients back to update embedding vectors. Besides deriving an upper-bound-based objective function from PageRank to represent the global node ranking. we also provide theoretical connection of the proposed proximity objective function to a general community detection solution. In sum, our model satisfies the following four model design characteristics: (I) *Scalability* [1, 6, 15, 18–21, 23, 26, 31]. We show that for each training epoch, our model enjoys linear time and space complexity to the number of nodes or links. Furthermore, different from some previous works relying on sampling non-existing links as negative examples for training, our model lifts the need to sample negative examples which not only saves extra training time but also relieves concern of sampling bias. (II) *Asymmetry* [2, 3, 15, 19, 20, 31]. Our model considers link directions to learn the embeddings of either directed or undirected networks. (III) *Unity* [1, 2, 6, 15, 18, 19, 21, 23, 24, 26, 31]. We perform joint learning to satisfy two different objective goals in a single model. The experiments show that the proposed multi-task neural network structure outperforms a two-stage model. (IV) *Simplicity*. Empirical verifications reflect that our model can achieve superior performance with only one hidden layer in neural networks and unified hyperparameter setting, freeing from fine-tuning the hyperparameters. This properly is especially important for an unsupervised learning task due to lack of validation data for fine-tuning. The source code of the proposed model can be downloaded here [6].

## 2 Related work

Recently, there exists growing number of works proposing embedding models specifically for network property preservation. Most of the prior methods extract latent embedding features by singular value decomposition or matrix factorization [1, 3, 8, 15, 19, 21, 22, 24, 28, 30]. Such methods typically define an $N$-by-$N$ matrix $\boldsymbol{A}$ ($N$ is the number of nodes) that reflect certain network properties, and then factorizes $\boldsymbol{A} \approx \boldsymbol{U}^{\top}\boldsymbol{V}$ or $\boldsymbol{A} \approx \boldsymbol{U}^{\top}\boldsymbol{U}$ into two low-dimensional embedding matrices $\boldsymbol{U}$ and $\boldsymbol{V}$.

There are also random-walk-based methods [6, 17, 18, 31] proposing an implicit reduction toward word embedding [14] by gathering random-walk sequences of sampled nodes throughout a network. The methods work well in practice but struggles to explain what network properties should be kept in their objective functions [20]. Unsupervised deep autoencoders are also used to learn latent embedding features of $\boldsymbol{A}$ [2, 23], especially achieve non-linear mapping strength through activation functions. Finally, some research defined different objective functions, like Kullback–Leibler divergence [20] or Huber loss [26] for network embedding. Please see Table 1 for detailed model comparisons.

Table 1: Model Comparisons. (I) Scalability; (II) Asymmetry; (III) Unity. No simplicity due to difficult comparisons between models with few sensitive and many insensitive hyperparameters.

| Model | (I) | (II) | (III) | Model | (I) | (II) | (III) |
|---|---|---|---|---|---|---|---|
| Proximity Embedding [19] | ✓ | ✓ | ✓ | MMDW [22] | | | ✓ |
| SocDim [21] | ✓ | | ✓ | SDNE [23] | ✓ | | ✓ |
| Graph Factorization [1] | ✓ | | ✓ | HOPE [15] | ✓ | ✓ | ✓ |
| DeepWalk [18] | ✓ | | ✓ | node2vec [6] | ✓ | | ✓ |
| TADW [28] | | ✓ | ✓ | HSCA [30] | | ✓ | ✓ |
| LINE [20] | ✓ | ✓ | | LANE [8] | | | ✓ |
| GraRep [3] | | | ✓ | APP [31] | ✓ | ✓ | ✓ |
| DNGR [2] | | ✓ | ✓ | M-NMF [24] | | | ✓ |
| TriDNR [17] | ✓ | | ✓ | NRCL [26] | ✓ | | ✓ |
| **Our PRUNE** | ✓ | ✓ | ✓ | | | | |

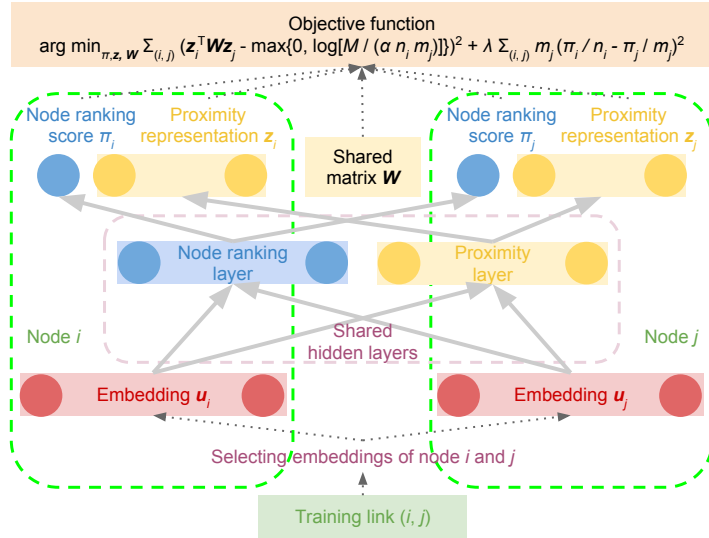

Figure 1: PRUNE overview. Each solid arrow represents a non-linear mapping function $h$ between two neural layers.

## 3 Model

### 3.1 Problem definition and notations

We are given a directed homogeneous graph or network $G = (V, E)$ as input, where $V$ is the set of vertices or nodes and $E$ is the set of directed edges or links. Let $N = |V|, M = |E|$ be the number of nodes and links in the network. For each node $i$, we denote $P_i, S_i$ respectively as the set of direct predecessors and successors of node $i$. Therefore, $m_i = |P_i|, n_i = |S_i|$ imply the in-degree and out-degree of the node $i$. Matrix $A$ denotes the corresponding adjacency matrix where each entry $a_{ij} \in [0, \infty)$ is the weight of link $(i, j)$. For simplicity, here we discuss only binary link weights: $a_{ij} = \{1, 0\}$ and $E = \{(i, j) : a_{ij} = 1\}$, but solutions for non-negative link weights can be derived in the same manner. Our goal is to build an unsupervised model, learning a $K$-dimensional embedding vector $u_i \in \mathbb{R}^K$ for each node $i$, such that $u_i$ preserves global node ranking and local proximity information.

### 3.2 Model overview

The Siamese neural network structure of our model is illustrated in Figure 1. Siamese architecture has been widely applied to multi-task learning like [27]. As Figure 1 illustrates, we define a pair of nodes $(i, j)$ as a training instance. Since both $i$ and $j$ refer to the same type of objects (i.e. nodes), it is natural to allow them to share the same hidden layer, which is what the Siamese architecture

suggests. We start from the bottom part in Figure 1 to introduce our proximity function. Here the model is trained using each link $(i, j)$ as a training instance. Given $(i, j)$, first our model feeds the existing embedding vectors $\boldsymbol{u}_i$ and $\boldsymbol{u}_j$ into the input layer. The values in $\boldsymbol{u}_i$ and $\boldsymbol{u}_j$ are updated by gradients propagated back from the output layers. To learn the mapping from the embedding vectors to objective functions, we put one hidden layer as bridge. Here we found the empirically one single hidden layers already yield competitive results, implying that a simple neural network is sufficient to encode graph properties into a vector space, which alleviates the burden on tuning hyperparameters in a neural network. Second, both nodes $i$ and $j$ share the same hidden layers in our neural networks, realizing by the Siamese neural networks. Each solid arrow in Figure 1 implies the following mapping function:

$$h(\boldsymbol{u}) = \phi(\boldsymbol{\omega}\boldsymbol{u} + \boldsymbol{b}) \tag{1}$$

where $\boldsymbol{\omega}, \boldsymbol{b}$ are the weight matrix and the bias vector. $\phi$ is an activation function leading to non-linear mappings. In Figure 1, our goal is to encode the proximity information in embedding space. Thus we define a $D$-dimensional vector $\boldsymbol{z} \in [0, \infty)^D$ that represents latent features of a node. In the next sections, we show that the proximity property can be modeled by the interaction between representations $\boldsymbol{z}_i$ and $\boldsymbol{z}_j$. We write down the mapping from embedding $\boldsymbol{u}$ to $\boldsymbol{z}$:

$$\boldsymbol{z} = \phi_2(\boldsymbol{\omega}_2 \phi_1(\boldsymbol{\omega}_1 \boldsymbol{u} + \boldsymbol{b}_1) + \boldsymbol{b}_2). \tag{2}$$

In Figure 1, we use the same network construction to encode an additional global node ranking $\pi \geq 0$. It is used to compare the relative ranks between one node and another. Formally, $\pi$ can be mapped from embedding $\boldsymbol{u}$ using the following formula:

$$\pi = \phi_4(\boldsymbol{\omega}_4 \phi_3(\boldsymbol{\omega}_3 \boldsymbol{u} + \boldsymbol{b}_3) + \boldsymbol{b}_4). \tag{3}$$

We impose the non-negative constraints of $\boldsymbol{z}, \pi$ for better theoretical property by exploiting the non-negative activation functions (ReLU or softplus for example) over the outputs $\phi_2$ and $\phi_4$. Other outputs of activation functions and all the $\boldsymbol{\omega}, \boldsymbol{b}$ are not limited to be non-negative. To add global node ranking information in proximity preservation, we construct a multi-task neural network structure as illustrated in Figure 1. Let the hidden layers for different network properties share the same embedding space. $\boldsymbol{u}$ is thus updated by the information simultaneously from multiple objective goals.

Different from a supervised learning task that the model can be trained by labeled data. Here instead we need to introduce an objective function for weight-tuning:

$$\underset{\pi \geq 0, \boldsymbol{z} \geq 0, \boldsymbol{W} \geq 0}{\arg \min} \sum_{(i,j) \in E} \left( \boldsymbol{z}_i^\top \boldsymbol{W} \boldsymbol{z}_j - \max \left\{ 0, \log \left( \frac{M}{\alpha m_j n_i} \right) \right\} \right)^2 + \lambda \sum_{(i,j) \in E} m_j \left( \frac{\pi_j}{m_j} - \frac{\pi_i}{n_i} \right)^2. \tag{4}$$

The first term aims at preserving the proximity and can be applied independently, as illustrated in Figure 1. The second term corresponds to the global node ranking task, which regularizes the relative scale among ranking scores. Here we import shared matrix $\boldsymbol{W} = \phi_5(\boldsymbol{\omega}_5)$ to learn the global linking correlations in the whole network. We also set non-negative-ranged activation function $\phi_5$ to satisfy non-negative $\boldsymbol{W}$. $\lambda$ controls the relative importance of these two terms. We will provide analysis for (4) in the next sections. Since the objective function (4) is differentiable, we are allowed to apply mini-batch stochastic gradient descent (SGD) to optimize every $\boldsymbol{\omega}, \boldsymbol{b}$ and even $u$ by propagating the gradients top-down from the output layers.

Deterministic mapping in (2) could be misunderstood that both $u$ and $z$ capture the same embedding information, but $z$ specifically captures the proximity property of a network through performing link prediction, and $u$ in fact tries to influence both proximity and global ranking. The reason to use $z$ instead of $u$ for link prediction is that we believe node ranking and link prediction are two naturally different tasks (but their information can be shared since highly ranked nodes can have better connectivity to others), using one single embedding representation $u$ to achieve both goals can lead to a compromised solution. Instead, $z$ can be treated as some "distilled" information extracted from $u$ specifically for link prediction, which can prevent our model from settling to a mediocre $u$ that fails to satisfy both goals directly.

### 3.3 Proximity preservation as PMI matrix tri-factorization

The first term in (4) aims at preserving the proximity property from input networks. We focus on the first-order and second-order proximity, which are explicitly addressed in several proximity-based

methods [3, 20, 23, 24]. The first-order proximity refers to whether node pair $(i, j)$ is connected in unweighted graphs. In an input network, links $(i, j) \in E$ are observed as positive training examples $a_{ij} = 1$. Thus, their latent inner product $\boldsymbol{z}_i^\top \boldsymbol{W} \boldsymbol{z}_j$ should be increased to reflect such close linking relationship. Nonetheless, usually another set of randomly chosen node pairs $(i, k) \in F$ is required to train the embedding model as negative training examples. Since set $F$ does not exist in input networks, one can sample $\alpha$ target nodes $k$ (with probability proportional to in-degree $m_k$) to form negative examples $(i, k)$. That is, given source node $i$, we emphasize the existence of link $(i, j)$ by distinguishing whether the corresponding target node is observed $((i, j) \in E)$ or not $((i, k) \in F)$. We can construct a binary logistic regression model to distinguish $E$ and $F$:

$$\arg\max_{\boldsymbol{z}, \boldsymbol{W}} \mathbb{E}_{(i,j) \in E} \left[ \log \sigma(\boldsymbol{z}_i^\top \boldsymbol{W} \boldsymbol{z}_j) \right] + \alpha \mathbb{E}_{(i,k) \in F} \left[ \log \left( 1 - \sigma(\boldsymbol{z}_i^\top \boldsymbol{W} \boldsymbol{z}_k) \right) \right] \tag{5}$$

where $\mathbb{E}$ denotes an expected value, $\sigma(x) = \frac{1}{1+\exp(-x)}$ is the sigmoid function. Inspired by the derivations in [12], we have the following conclusion:

**Lemma 3.1.** *Let $y_{ij} = \boldsymbol{z}_i^\top \boldsymbol{W} \boldsymbol{z}_j$. We have the closed-form solution from zero first-order derivative of (5) over $y_{ij}$:*

$$y_{ij} = \log \frac{M}{\alpha n_i m_j} = \log \frac{p_{s,t}(i,j)}{p_s(i) p_t(j)} - \log \alpha \tag{6}$$

*where $p_{s,t}(i,j) = \frac{1}{|E|} = \frac{1}{M}$ is the joint probability of link (positive example) $(i,j)$ in set $E$, $p_s(i) = \frac{n_i}{M}$ follows a distribution proportional to out-degree $n_i$ of source node $i$, whereas $p_t(j) = \frac{m_j}{M}$ follows another distribution proportional to in-degree $m_j$ of target node $j$.*

*Proof.* Please refer to our Supplementary Material Section 2. $\square$

Clearly, (6) is the pointwise mutual information (PMI) shifted by $\log \alpha$, which can be viewed as link weights in terms of out-degree $n_i$ and in-degree $m_j$. If we directly minimize the difference between two sides in (6) rather than maximize (5), then we are free from sampling negative examples $(i, k)$ to train a model. Following the suggestions in [12], we filter negative (less informative) PMI as shown in (4), causing further performance improvement.

The second-order proximity refers to the fact that the similarity of $\boldsymbol{z}_i$ and $\boldsymbol{z}_j$ is higher if nodes $i, j$ have similar sets of direct predecessors and successors (that is, the similarity reflects 2-hop distance relationships). Now we present how to preserve the second-order proximity using tri-factorization-based link prediction [13, 32]. Let $\boldsymbol{A}^{\text{PMI}} = \left[ \max \left\{ 0, \log \frac{M}{\alpha n_i m_j} \right\} \text{ if } (i, j) \in E; \text{ otherwise missing} \right]$ be the corresponding PMI matrix. Link prediction aims to predict the missing PMI values in $\boldsymbol{A}^{\text{PMI}}$. Factorization methods suppose $\boldsymbol{A}^{\text{PMI}}$ of low-rank $D$, and then learn matrix tri-factorization $\boldsymbol{Z}^\top \boldsymbol{W} \boldsymbol{Z} \approx \boldsymbol{A}^{\text{PMI}}$ using non-missing entries. Matrix $\boldsymbol{Z} = [\boldsymbol{z}_1 \boldsymbol{z}_2 \dots \boldsymbol{z}_N]$ aligns latent representations with link distributions. Compared with classical factorization $\boldsymbol{Z}^\top \boldsymbol{V}$, such tri-factorization supports the asymmetric transitivity property of directed links. Specifically, the existence of two directed links $(i, j)$ $(\boldsymbol{z}_i^\top \boldsymbol{W} \boldsymbol{z}_j)$, $(j, k)$ $(\boldsymbol{z}_j^\top \boldsymbol{W} \boldsymbol{z}_k)$ increase the likelihood of $(i, k)$ $(\boldsymbol{z}_i^\top \boldsymbol{W} \boldsymbol{z}_k)$ via representation propagation $\boldsymbol{z}_i \to \boldsymbol{z}_j \to \boldsymbol{z}_k$, but not the case for $(k, i)$ due to asymmetric $\boldsymbol{W}$. Then we have a lemma as follows:

**Lemma 3.2.** *Matrix tri-factorization $\boldsymbol{Z}^\top \boldsymbol{W} \boldsymbol{Z} \approx \boldsymbol{A}^{PMI}$ preserves the second-order proximity.*

*Proof.* Please refer to our Supplementary Material Section 3. $\square$

Next, we discuss the connection between matrix tri-factorization and community. Different from heuristic statements in [13, 32], we argue that the representation vector $\boldsymbol{z}_i$ captures a $D$-community distribution for node $i$ (each dimension is proportional to the probability that node $i$ belongs to certain community), and shared matrix $\boldsymbol{W}$ implies the interactions among these $D$ communities.

**Lemma 3.3.** *Matrix tri-factorization $\boldsymbol{z}_i^\top \boldsymbol{W} \boldsymbol{z}_j$ can be regarded as the expectation of community interactions with distributions of link $(i, j)$.*

$$\boldsymbol{z}_i^\top \boldsymbol{W} \boldsymbol{z}_j \propto \mathbb{E}_{(i,j)} [\boldsymbol{W}] = \sum_{c=1}^{D} \sum_{d=1}^{D} \Pr(i \in C_c) \Pr(j \in C_d) w_{cd}, \tag{7}$$

*where each entry $w_{cd}$ is the expected number of interactions from community $c$ to $d$, and $C_c$ denotes the set of nodes in community c.*

*Proof.* Please refer to the Supplementary Material Section 4. ∎

Based on the binary classification model (5), when a true link $(i,j)$ is observed in the training data, the corresponding inner product $z_i^\top W z_j$ is increased, which is equivalent to raising the expectation $\mathbb{E}_{(i,j)}[W]$.

To summarize, the derivations from logistic classification (5) to PMI matrix tri-factorization (6) show the tri-factorization model preserves the first-order proximity. Then Lemma 3.2 proves the preservation of second-order proximity. Besides, if a non-negative constraint is imposed, Lemma 3.3 shows that the tri-factorization model can be interpreted as capturing community interactions. That says, our proximity preserving loss achieves the first-order proximity, second-order proximity, and community preservation.

Given non-negative $\log \frac{M}{n_i m_j}$ as our setting in (4), we make another observation on community detection. (6) can be rewritten as the following equation:

$$\underbrace{1 - \exp\left(-z_i^\top W z_j\right)}_{\mathcal{P}(X^{(i,j)}>0)=1-\mathcal{P}(X^{(i,j)}=0)} = \underbrace{1 - \frac{n_i m_j}{M}}_{\text{Modularity as } a_{ij}=1, \alpha=1}. \tag{8}$$

Following Lemma 3.3, we can then derive

**Lemma 3.4.** *The left-hand side of (8) is the probability $\mathcal{P}(X^{(i,j)} > 0)$, where $0 \le X^{(i,j)} \le D^2$ represents the total numbers of interactions between all the community pairs $(c,d) \ \forall 1 \le c \le D, 1 \le d \le D$ that affect the existence of this link $(i,j)$, following Poisson distribution $\mathcal{P}(X^{(i,j)})$ with mean $z_i^\top W z_j$.*

*Proof.* Please refer to the Supplementary Material Section 5. ∎

In fact, on either side of Equation (8), it evaluates the likelihood of the occurrence of a link. For the left-hand side, as shown in reference [29] and our Supplementary Material 5, an existing link implies at least one community interactions $(X > 0)$, whose probability is assumed following Poisson with means equal to the tri-factorization values. The right-hand side is commonly regarded as the "modularity" [11], which measures the difference between links from the observed data and links from random generation. Modularity is commonly used as an evaluation metric for the quality of a community detection algorithm (see [21, 24]). The deep investigation of Equation (8) is left for our future work.

### 3.4 Global node ranking preservation as PageRank upper bound

Here we want to connect the second objective to PageRank. To be more precise, the second term in (4) (without parameter $\lambda$) comes from an upper bound of PageRank assumption. PageRank [16] is arguably the most common unsupervised method to evaluate the rank of a node. It claims that ranking score of a node $j$ $\pi_j$ is the probability of visiting $j$ through random walks. $\pi_j \ \forall \ j \in V$ can be obtained from the ranking score accumulation from direct predecessors $i$, weighted by the reciprocal of out-degree $n_i$. One can express PageRank using the minimization of squared loss $L = \sum_{j \in V} (\sum_{i \in P_j} \frac{\pi_i}{n_i} - \pi_j)^2$. Here the probability constraint $\sum_{i \in V} \pi_i = 1$ is not considered since we care only about the relative rankings. The damping factor in PageRank is not considered either for model simplicity. Unfortunately, it is infeasible to apply SGD to update $L$, since summation $\sum_{i \in P_j}$ is inside the square, violating the standard SGD assumption $L = \sum_{(i,j) \in E} L_{ij}$ where each sub-objective function $L_{ij}$ is relevant to a single training link $(i,j)$. Instead, we choose to minimize an upper bound.

**Lemma 3.5.** *By Cauchy–Schwarz inequality, we have the upper bound as follows:*

$$\sum_{j \in V} \left( \sum_{i \in P_j} \frac{\pi_i}{n_i} - \pi_j \right)^2 \le \sum_{(i,j) \in E} m_j \left( \frac{\pi_i}{n_i} - \frac{\pi_j}{m_j} \right)^2. \tag{9}$$

*Proof.* Please refer to our Supplementary Material Section 6. □

The proof of approximation ratio of such upper bound (9) is left as our future work. Nevertheless, as will be shown later, the experiments have demonstrated the effectiveness of such upper bound. Intuitively, (9) minimizes the difference between $\frac{\pi_i}{n_i}$ and $\frac{\pi_j}{m_j}$ weighted by in-degree $m_j$. This could be explained by the following lemma:

**Lemma 3.6.** *The objectvie $\frac{\pi_i}{n_i} = \frac{\pi_j}{m_j}$ at the right-hand side of (9) is a sufficient condition of the objective $\sum_{i \in P_j} \frac{\pi_i}{n_i} = \pi_j$ at the left-hand side of (9).*

*Proof.* Please refer to our Supplementary Material Section 7. □

### 3.5 Discussion

We have mentioned four major advantages of our model the introduction section. Here we would like to provide in-depth discussions on them. (I) *Scalability*. Since only the positive links are used for training, during SGD, our model spends $O(M\Omega^2)$ time for each epoch, where $\Omega$ is the maximum number of neurons of a layer in our model, which is usually in the hundreds. Also, our model costs only $O(N + M)$ space to store input networks and the sparse PMI matrix consumes $O(M)$ non-zero entries. In practice $\Omega^2 \ll M$, our model is thus scalable. (II) *Asymmetry*. By the observation in (4), replacing $(i, j)$ with $(j, i)$ leads to different results since $\boldsymbol{W}$ and PageRank upper bound are asymmetric. (III) *Unity*. All the objectives in our model are jointly optimized under a multi-task Siamese neural network. (IV) *Simplicity*. As experiments shows, our model performs well with single hidden layers and the same hyperparameter setting across all the datasets, which could alleviate the difficult hyperparameter determination for unsupervised network embedding.

## 4 Experiments

### 4.1 Settings

**Datasets.** We benchmark our model on three real-world networks in different application domains:

(I) *Hep-Ph* [7]. It is a paper citation network from 1993 to 2003, including $34,546$ papers and $421,578$ citations relationships. Following the same setup as [25], we leave citations before 1999 for embedding generation, and then evaluate paper ranks using the number of citations after 2000.

(II) *Webspam* [8]. It is a web page network used in Webspam Challenges. There are $114,529$ web pages and $1,836,441$ hyperlinks. Participants are challenged to build a model to rank the $1,933$ labeled non-spam web pages higher than $122$ labeled spam ones.

(III) *FB Wall Post* [9]. Previous task [7] aims at ranking active users using a $63,731$-user, $831,401$-link wall post network in social media website Facebook, New Orlean 2009. The nodes denote users and a link implies that a user posts at least an article on someone's wall. $14,862$ users are marked active, that is, they continue to post articles in the next three weeks after a certain date. The goal is to rank active users over inactive ones.

**Competitors.** We compare the performance of our model with DeepWalk [18], LINE [20], node2vec [6], SDNE [23] and NRCL [26]. DeepWalk, LINE and node2vec are popular models used in various applications. SDNE proposes another neural network structure to embed networks. NRCL is one of the state-of-the-art network embedding model, specially designed for link prediction. Note that NRCL encodes external node attributes into network embedding, but we discard this part since such information are not assumed available in our setup.

**Model Setup.** For all experiments, our model fixes node embedding and hidden layers to be 128-dimensional, proximity representation to be 64-dimensional. Exponential Linear Unit (ELU) [4] activation is adopted in hidden layers for faster learning, while output layers use softplus activation for node ranking score and Rectified Linear Unit (ReLU) [5] activation for proximity representation

to avoid negative-or-zero scores as well as negative representation values. We recommend and fix $\alpha = 5$, $\lambda = 0.01$. All training uses a batch size of 1024 and Adam [9] optimizer with learning rate 0.0001.

**Evaluation.** Similar to the previous works, we want to evaluate our embedding using supervised learning tasks. That is, we want to evaluate whether the proposed embedding yields better results for a (1) learning-to-rank (2) classification and regression (3) link prediction tasks.

## 4.2 Results

In the following paragraphs, we call our proposed model *PRUNE*. *PRUNE* without the global ranking part is named *TriFac* below.

**Learning-to-rank.** In this setting, we use pairwise approach that formulates learning-to-rank as a binary classification problem and take embeddings as node attributes. Linear Support Vector Machine with regularization $C = 1.0$ is used as our learning-to-rank classifier. We train on $80\%$ and evaluate on $20\%$ of datasets. Since Webspam and FB Wall Post possess binary labels, we choose Area Under ROC Curve (AUC) as the evaluation metric. Following the setting in [25], Hep-Ph paper citation is a real value, and thus suits better for Spearman's rank correlation coefficient.

The results in Table 2 show that *PRUNE* significantly outperforms the competitors. Note that *PRUNE* which incorporates global node ranking as a multi-task learning has superior performance compared with *TriFac* which only considers the proximity. It shows that the unsupervised global ranking we modeled is positively correlated with the rankings in these learning-to-ranking tasks. Also the multi-task learning enriches the underlying interactions between two tasks and is the key to better performance of *PRUNE*.

Table 2: Learning-to-rank performance (†: outperforms 2nd-best with p-value < 0.01).

| Dataset | Evaluation | DeepWalk | LINE | node2vec | SDNE | NRCL | TriFac | PRUNE |
|---------|-----------|----------|------|----------|------|------|--------|-------|
| Hep-Ph | Rank Corr. | 0.485 | 0.430 | 0.494 | 0.353 | 0.327 | 0.554 | **0.621**† |
| Webspam | AUC | 0.821 | 0.818 | 0.843 | 0.800 | 0.839 | 0.821 | **0.853**† |
| FB Wall Post | AUC | 0.702 | 0.712 | 0.730 | 0.749 | 0.573 | 0.747 | **0.765**† |

**Classification and Regression.** In this experiment, embedding outputs are directly used for binary node classification on Webspam and FB Wall Post and node regression on Hep-Ph. We only observe $80\%$ nodes while training and predict the labels of remaining $20\%$ nodes. Random Forest and Support Vector Regression are used for classification and regression, respectively. Classification is evaluated by AUC and regression is evaluated by the Root Mean Square Error (RMSE). Table 3 shows that *PRUNE* reaches the lowest RMSE on the regression task and the highest AUC on two classification tasks among embedding algorithms, while *TriFac* is competitive to others. The results show that the global ranking modeled by us contains useful information to capture certain properties of nodes.

Table 3: Classification and regression performance (†: outperforms 2nd-best with p-value < 0.01).

| Dataset | Evaluation | DeepWalk | LINE | node2vec | SDNE | NRCL | TriFac | PRUNE |
|---------|-----------|----------|------|----------|------|------|--------|-------|
| Hep-Ph | RMSE | 12.079 | 12.307 | 11.909 | 12.451 | 12.429 | 11.967 | **11.720**† |
| Webspam | AUC | 0.620 | 0.597 | 0.622 | 0.605 | 0.578 | 0.576 | **0.637**† |
| FB Wall Post | AUC | 0.733 | 0.707 | 0.744 | 0.752 | 0.759 | 0.763 | **0.775**† |

**Link Prediction.** We randomly split network edges into $80\%$-$20\%$ train-test subsets as positive examples and sample equal number of node pairs with no edge connection as negative samples. Embeddings are learned on the training set and performance is evaluated on the test set. Logistic regression is adopted as the link prediction algorithm and models are evaluated by AUC. The results in Table 4 show that *PRUNE* outperforms all counterparts significantly, while *TriFac* is competitive to others. The results, together with previous two experiments, demonstrate the effectiveness of *PRUNE* for diverse network applications.

**Robustness to Noisy Data.** In the real-world settings, usually only partial network is observable as links can be missing. Perturbation analysis is then conducted in verifying the robustness of models by measuring the learning-to-rank performance when different fractions of edges are missing. Figure 2 shows that *PRUNE* persistently outperforms competitors across different fractions of missing

Table 4: Link prediction performance (†: outperforms 2nd-best with p-value < 0.01).

| Dataset | DeepWalk | LINE | node2vec | SDNE | NRCL | TriFac | PRUNE |
|---------|----------|------|----------|------|------|--------|-------|
| Hep-Ph | 0.803 | 0.796 | 0.805 | 0.751 | 0.688 | 0.814 | **0.861**† |
| Webspam | 0.885 | 0.954 | 0.894 | 0.953 | 0.910 | 0.946 | **0.973**† |
| FB Wall Post | 0.828 | 0.781 | 0.853 | 0.855 | 0.731 | 0.858 | **0.878**† |

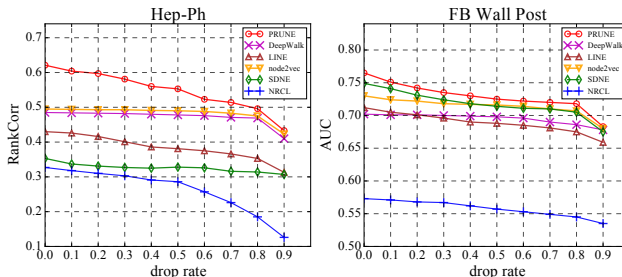

Figure 2: Perturbation analysis for learning-to-rank on Hep-Ph and FB Wall Post.

edges. The results demonstrate its robustness to missing edges which is crucial for evolving or costly-constructed networks.

**Discussions.** The superiority can be summarized based on the features of the models:

(I) We have an explicit objective to optimize. Random walk based models (i.e. DeepWalk, node2vec) lack such objectives and moreover, noises are introduced during the random walk procedure.

(II) We are the only model that considers global node ranking information.

(III) We preserve first and second-order proximity and considers the asymmetry (i.e. direction of links). NRCL only preserves the first-order proximity and does not consider asymmetry. SDNE does not consider asymmetry either. LINE does not handle first-order and second-order proximity jointly but instead treating them independently.

## 5   Conclusion

We propose a multi-task Siamese deep neural network to generate network embeddings that preserve global node ranking and community-aware proximity. We design a novel objective function for embedding training and provide corresponding theoretical interpretation. The experiments shows that preserving the properties we have proposed can indeed improve the performance of supervised learning tasks using the embedding as features.

### Acknowledgments

This study was supported in part by the Ministry of Science and Technology (MOST) of Taiwan, R.O.C., under Contracts 105-2628-E-001-002-MY2, 106-2628-E-006-005-MY3, 104-2628-E-002 -015 -MY3 & 106-2218-E-002 -014 -MY4 , Air Force Office of Scientific Research, Asian Office of Aerospace Research and Development (AOARD) under award number No.FA2386-17-1-4038, and Microsoft under Contracts FY16-RES-THEME-021. All opinions, findings, conclusions, and recommendations in this paper are those of the authors and do not necessarily reflect the views of the funding agencies.

## Footnotes

[4] http://webspam.lip6.fr/wiki/pmwiki.php

[5] https://wsdmcupchallenge.azurewebsites.net/

[6] https://github.com/ntumslab/PRUNE

[7] http://snap.stanford.edu/data/cit-HepPh.html

[8] http://chato.cl/webspam/datasets/uk2007/

[9] http://socialnetworks.mpi-sws.org/data-wosn2009.html

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
