[Supplementary Material · PRUNE_Supplementary_Material_NIPS.pdf]

# PRUNE: Preserving Proximity and Global Ranking for Network Embedding (Supplementary Material)

**Yi-An Lai** [*‡]
National Taiwan University
b99202031@ntu.edu.tw

**Chin-Chi Hsu** [†‡]
Academia Sinica
chinchi@iis.sinica.edu.tw

**Wen-Hao Chen** [*]
National Taiwan University
b02902023@ntu.edu.tw

**Mi-Yen Yeh** [†]
Academia Sinica
miyen@iis.sinica.edu.tw

**Shou-De Lin** [*]
National Taiwan University
sdlin@csie.ntu.edu.tw

# 1 Notation introduction

Table 1: Commonly used notations

| Notation | Description |
|---|---|
| $G = (V, E)$ | Input directed network (or graph) |
| $\boldsymbol{A} \in \{0,1\}^{N \times N}$ | Adjacency matrix of network $G$ |
| $V$ | Set of nodes or vertices |
| $E = \{(i,j) : a_{ij} = 1\}$ | Set of links or edges |
| $N = \|V\|$ | Number of nodes in network $G$ |
| $M = \|E\|$ | Number of links in network $G$ |
| $P_i$ | Set of direct predecessors of node $i$ |
| $S_i$ | Set of direct successors of node $i$ |
| $m_i = \|P_i\|$ | In-degree of node $i$ |
| $n_i = \|S_i\|$ | Out-degree of node $i$ |
| $\boldsymbol{z}_i \in [0, \infty)^D$ | Latent $D$-community distribution vector of node $i$ |
| $\boldsymbol{W} \in [0, \infty)^{D \times D}$ | Shared matrix of community interactions |
| $\pi_i \geq 0$ | Global ranking score of node $i$ |

[*]Department of Computer Science and Information Engineering
[†]Institute of Information Science
[‡]These authors contributed equally to this paper.

## 2 Proof for the closed-form solution of binary classification

The objective function of our binary classification is shown below:

$$\underset{\boldsymbol{z},\boldsymbol{W}}{\arg\max}\mathbb{E}_{(i,j)\in E}\left[\log\sigma(\boldsymbol{z}_i^\top\boldsymbol{W}\boldsymbol{z}_j)\right]+\alpha\mathbb{E}_{(i,j)\in F}\left[\log\left(1-\sigma(\boldsymbol{z}_i^\top\boldsymbol{W}\boldsymbol{z}_k)\right)\right]$$

$$=\mathbb{E}_i\mathbb{E}_{j\in S_i}\left[\log\sigma(\boldsymbol{z}_i^\top\boldsymbol{W}\boldsymbol{z}_j)\right]+\alpha\mathbb{E}_i\mathbb{E}_k\left[\log\left(1-\sigma(\boldsymbol{z}_i^\top\boldsymbol{W}\boldsymbol{z}_k)\right)\right]$$

$$=\sum_{i\in V}\sum_{j\in S_i}p_s(i)p_t(j|i)\log\sigma(\boldsymbol{z}_i^\top\boldsymbol{W}\boldsymbol{z}_j)+\alpha\sum_{i\in V}\sum_{k\in V}p_s(i)p_t(k)\log\left(1-\sigma(\boldsymbol{z}_i^\top\boldsymbol{W}\boldsymbol{z}_k)\right)$$

$$=\sum_{i\in V}\sum_{j\in S_i}\frac{n_i}{M}\frac{1}{n_i}\log\sigma(\boldsymbol{z}_i^\top\boldsymbol{W}\boldsymbol{z}_j)+\alpha\sum_{i\in V}\sum_{k\in V}\frac{n_i}{M}\frac{m_k}{M}\log\left(1-\sigma(\boldsymbol{z}_i^\top\boldsymbol{W}\boldsymbol{z}_k)\right).$$

Given source node $i$, one of linked target node $j \in S_i$ enjoys a conditional distribution proportional to $\frac{1}{n_i}$. Since $S_i \subseteq V$ implies $k$ including $j$, for specific positive example $(i,j)$, we have:

$$\arg\max L_{ij}=\frac{1}{M}\log\sigma(\boldsymbol{z}_i^\top\boldsymbol{W}\boldsymbol{z}_j)+\alpha\frac{n_i}{M}\frac{m_j}{M}\log\left(1-\sigma(\boldsymbol{z}_i^\top\boldsymbol{W}\boldsymbol{z}_j)\right).$$

Now let $y_{ij}=\boldsymbol{z}_i^\top\boldsymbol{W}\boldsymbol{z}_j$. We first derive the closed-form solution of zero first-order derivative over $\sigma(y_{ij})$:

$$\frac{\partial L_{ij}}{\partial\sigma(y_{ij})}=\frac{1}{M}\frac{1}{\sigma(y_{ij})}-\alpha\frac{n_i}{M}\frac{m_j}{M}\frac{1}{1-\sigma(y_{ij})}$$

$$=0$$

$$\implies\sigma(y_{ij})=\frac{\frac{1}{M}}{\frac{1}{M}+\alpha\frac{n_i}{M}\frac{m_j}{M}}$$

$$=\frac{M}{M+\alpha n_i m_j}.$$

Next We obtain $y_{ij}$ after calculations:

$$\frac{1}{1+e^{-y_{ij}}}=\frac{M}{M+\alpha n_i m_j}$$

$$\implies y_{ij}=\log\frac{M}{\alpha n_i m_j}$$

$$=\log\frac{\frac{1}{M}}{\alpha\frac{n_i}{M}\frac{m_j}{M}}$$

$$=\log\frac{p_{s,t}(i,j)}{p_s(i)p_t(j)}-\log\alpha.$$

## 3 Proof for matrix tri-factorization supporting the second-order proximity

The second-order proximity implies high similarity between two representation vectors $\boldsymbol{z}_i, \boldsymbol{z}_j$ if nodes $i, j$ have similar sets of direct predecessors or direct successors.

Consider the non-missing entries of the $i$-th and $j$-th column $\boldsymbol{a}_i^{\mathrm{PMI}}, \boldsymbol{a}_j^{\mathrm{PMI}}$ in our derived PMI matrix $\boldsymbol{A}^{\mathrm{PMI}}$. Since all the non-missing entries are in link set $E$, the two columns represent the sets of direct predecessors of node $i$ and $j$ where the links are weighted by PMI. Based on our matrix tri-factorization $\boldsymbol{Z}^\top\boldsymbol{W}\boldsymbol{Z}\approx\boldsymbol{A}^{\mathrm{PMI}}$, we have:

$$\boldsymbol{a}_i^{\mathrm{PMI}}\approx\boldsymbol{Z}^\top\boldsymbol{W}\boldsymbol{z}_i,$$

$$\boldsymbol{a}_j^{\mathrm{PMI}}\approx\boldsymbol{Z}^\top\boldsymbol{W}\boldsymbol{z}_j$$

where $\boldsymbol{z}_i$ is the $i$-th column of representation matrix $\boldsymbol{Z}$. As the predecessor sets are similar $\boldsymbol{a}_i^{\mathrm{PMI}}\approx \boldsymbol{a}_j^{\mathrm{PMI}}$, then their corresponding representation vector must be similar $\boldsymbol{z}_i\approx\boldsymbol{z}_j$ due to the same weight matrix $\boldsymbol{Z}^\top\boldsymbol{W}$. Similarly, when modeling the matrix tri-factorization for the rows in $\boldsymbol{A}^{\mathrm{PMI}}$, we also obtain $\boldsymbol{z}_i\approx\boldsymbol{z}_j$ if nodes $i, j$ have similar successor sets.

# 4  Proof for the expectation of community interactions

Let $\boldsymbol{W} \in [0, \infty)^{D \times D}$ be the community interaction matrix where each entry $w_{cd}$ denotes the expected number of interactions from community $c$ to $d$. $c = d$ implies the number of internal interactions within a community. We assume that the existence of link $(i, j)$ is determined by the expected value of $\boldsymbol{W}$ with community distributions of $i$ and $j$:

$$\mathbb{E}_{(i,j)}[\boldsymbol{W}] = \sum_{c=1}^{D} \sum_{d=1}^{D} \Pr(i \in C_c, j \in C_d) w_{cd}$$

where $C_c$ is the set of nodes in community $c$. Let $\boldsymbol{z}_i$ be an unnormalized distribution vector where each dimension $0 \le z_{ic} \propto \Pr(i \in C_c)$. Under the independence assumption between $\Pr(i \in C_c)$ and $\Pr(j \in C_d)$, we have:

$$\sum_{c=1}^{D} \sum_{d=1}^{D} \Pr(i \in C_c, j \in C_d) w_{cd} = \sum_{c=1}^{D} \sum_{d=1}^{D} \Pr(i \in C_c) \Pr(j \in C_d) w_{cd}$$
$$\propto \sum_{c=1}^{D} \sum_{d=1}^{D} z_{ic} z_{jd} w_{cd}$$
$$= \boldsymbol{z}_i^\top \boldsymbol{W} \boldsymbol{z}_j.$$

# 5  Proof for community interactions following Poisson distribution

Based on the proof in the previous section, for specific link $(i, j)$, the expected number of interactions from community $c$ to $d$ is

$$\Pr(i \in C_c) \Pr(j \in C_d) w_{cd} \propto z_{ic} z_{jd} w_{cd}.$$

Here we model discrete random variable $X_{cd}^{(i,j)}$ as the number of interactions from community $c$ to $d$ for link $(i, j)$, following Poisson distribution $X_{cd}^{(i,j)} \sim \mathcal{P}(\mu = z_{ic} z_{jd} w_{cd})$. Using the properties of Poisson distribution, the overall number of interactions among community pairs is

$$X^{(i,j)} = \sum_{c=1}^{D} \sum_{d=1}^{D} X_{cd}^{(i,j)} \sim \mathcal{P}\left(\mu = \sum_{c=1}^{D} \sum_{d=1}^{D} z_{ic} z_{jd} w_{cd} = \boldsymbol{z}_i^\top \boldsymbol{W} \boldsymbol{z}_j\right).$$

Assume that node $i$ and $j$ belong to at least one community. Link $(i, j)$ exists due to at least one interaction between the communities that $i$ and $j$ belong to, which is

$$\mathcal{P}(X^{(i,j)} > 0) = 1 - \mathcal{P}(X^{(i,j)} = 0) = 1 - \exp(-\boldsymbol{z}_i^\top \boldsymbol{W} \boldsymbol{z}_j).$$

# 6 Proof for PageRank upper-bound objective function

Let $P_j$ be the set of direct predecessors of node $j$, and $n_i$ be the out-degree of node $i$. Then we have:

$$\operatorname*{arg\,min}_{\pi} \sum_{j \in V} \left( \sum_{i \in P_j} \frac{\pi_i}{n_i} - \pi_j \right)^2 = \sum_{j \in V} \left( \left( \sum_{i \in P_j} \frac{\pi_i}{n_i} \right)^2 - 2\pi_j \sum_{i \in P_j} \frac{\pi_i}{n_i} + \pi_j^2 \right)$$

$$\leq \sum_{j \in V} \left( \underbrace{\left( \sum_{i \in P_j} 1^2 \right) \left( \sum_{i \in P_j} \left( \frac{\pi_i}{n_i} \right)^2 \right)}_{\text{Cauchy–Schwarz inequality}} - 2\pi_j \sum_{i \in P_j} \frac{\pi_i}{n_i} + \pi_j^2 \right)$$

$$= \sum_{j \in V} \sum_{i \in P_j} \left( m_j \left( \frac{\pi_i}{n_i} \right)^2 - 2\pi_j \frac{\pi_i}{n_i} + \frac{1}{m_j}\pi_j^2 \right)$$

$$= \underbrace{\sum_{(i,j) \in E}}_{=j \in V, i \in P_j} m_j \left( \left( \frac{\pi_i}{n_i} \right)^2 - 2\frac{\pi_j \pi_i}{m_j n_i} + \left( \frac{\pi_j}{m_j} \right)^2 \right)$$

$$= \sum_{(i,j) \in E} m_j \left( \frac{\pi_i}{n_i} - \frac{\pi_j}{m_j} \right)^2 .$$

Since $\left( \sum_{i \in P_j} 1^2 \right) \left( \sum_{i \in P_j} \left( \frac{\pi_i}{n_i} \right)^2 \right) \geq 0$, we constrain $\pi_i \geq 0$ for all node $i$ to make the upper bound tighter.

# 7 Proof for PageRank sufficient condition

For each node $j \in V$, let $P_j$ be the set of direct predecessors of node $j$. We denote node $i \in P_j$. Then for each node $j$, we show a sufficient condition:

$$\frac{\pi_i}{n_i} = \frac{\pi_j}{m_j} \ \forall \ \underbrace{i \in P_j, j \in V}_{=(i,j) \in E}$$

where $m_j = |P_j|$, $n_i$ is respectively the in-degree of node $j$ and the out-degree of node $i$. Now we calculate the sum of the left-hand-side for all the direct predecessors $i$ of each node $j$:

$$\sum_{i \in P_j} \frac{\pi_i}{n_i} = \sum_{i \in P_j} \frac{\pi_j}{m_j}$$

$$= \frac{1}{m_j} \sum_{i \in P_j} \pi_j$$

$$= \frac{1}{m_j} m_j \pi_j$$

$$= \pi_j \ \forall \ j \in V.$$

The equation is just the PageRank assumption: $\sum_{i \in P_j} \frac{\pi_i}{n_i} = \pi_j \ \forall \ j \in V$ (here we omit the damping factor).