[Reviews · NeurIPS 2017]

Reviewer 1



This paper proposes an unsupervised network embedding learning framework. The contributions include: a) designing an objective function based on matrix tri-factorization which can preserve the proximity and the global node ranking of the graph simultaneously; b) providing sufficient theoretical derivations to the objective function they designed; c) testing the performance of node embeddings, which are learned by optimizing the objective function through a neural network, on three real world datasets and several data mining tasks. Pros: +: This paper has good theoretical foundations, and the loss function can be well explained from the proximity preserving and global ranking preserving view. +: The results are fairly strong on all the tasks and datasets. Cons: -: The paper puts too much space on theoretical derivations but little on experiments, and some lemmas are trivial thus should be omitted or concisely written, e.g., Lemma 3.5 and Lemma 3.6. -: Part in Line 169 ~ Line 184 is not directly related to the final objective function. Maximizing the modularity is the special case of (6) when alpha = 1, and they don’t use this function as a part of the final loss function, thus I think this part is relatively irrelevant to the main framework. -: Also it seems quite straightforward to incorporate node ranking information in their frameworks. The simplest way is to add a second objective, like the way the authors did in this paper, although the experiment results show that adding this part boosts the performance. -: Some writing details should be improved. In the experiment part, results under different evaluation metrics should be listed separately. And I suggest to add one more baseline, i.e., GCN (Kipf and Welling (ICLR 2017)), which is a strong node embedding learning method based on graph convolution network.

Reviewer 2



The paper presents a NN model for learning graph embeddings that preserves the local graph structure and a global node ranking similar to PageRank. The model is based on a Siamese network, which takes as inputs two node embeddings and compute a new (output) representation for each node using the Siamese architecture. Learning is unsupervised in the sense that it makes use only of the graph structure. The loss criterion optimizes a weighted inner product of the output representations (local loss) plus a node ranking criterion (global loss). Some links with a community detection criterion are also discussed. The model is evaluated on a series of tasks: node ranking, classification and regression, link prediction, and compared to other families of unsupervised embedding learning methods. The paper introduces new loss criteria and an original method for learning graph embeddings. The introduction of a node ranking term based on the graph structure is original. The authors find that this term brings interesting benefits to the learned embeddings and allows their method to outperform the other unsupervised baselines for a series of tasks. The links with the community criteria is also interesting. On the other hand, the organization and the form of the paper makes it somewhat difficult to follow at some places. The motivation for using a Siamese architecture is not explained. Two representations are learned for each node (notations u and z in the paper). Since the z representation is a deterministic function of the u representation, why are these two different representations needed, and why not directly optimizing the initial u representation? The explanation for the specific proximity preserving loss term is also unclear. For example, it is shown that this criterion optimizes a “second-order” proximity, but this term is never properly defined. Even if the result looks interesting, lemma 3.4 which expresses links with a community detection criterion is confusing. In the experimental section, a more detailed description of the baselines is needed in order to understand what makes them different from the proposed method. The latter outperforms all the baselines on all the tests, but there is no analysis of the reasons for this good behavior. Overall the paper reveals intriguing and interesting findings. The proposed method has several original aspects, but the paper requires a better organization/presentation in order to fully appreciate the proposed ideas.

Reviewer 3



This paper studied the network embedding problem, with focuses on proximity and global ranking preserve. The authors proposed an unsupervised Siamese neural network structure to learn node embedding following a generative fashion. They claimed that their model satisfies four characteristics, which are scalability, asymmetry, unity and simplicity.   The Siamese neural network is designed as a multi-task learning framework, with shared hidden layers for embedding vectors of each pair of connected nodes. The node ranking hidden layer is designed to encode the global ranking information, and the proximity hidden layer is designed for preserving local proximity information. As a result, this generative framework updates the two input embedding vectors using gradients propagated back from the output layers on an objective function consists of two parts of rewards. one for each task. The network design looks simple and robust with only one hidden layer, that leads to much fewer hyper-parameters to tune. And the authors provide detailed proof for second-order proximity preservation, and proof for global ranking preservation with a upper bound of PageRank.  This paper is well-written and easy to follow. Figure 1 is nice and clean, which provides a short cut for reader to understand their model.  The experimental part is well-designed, the results demonstrated that the proposed embedding model outperforms its competitors on task like rank prediction, classification/regression and link prediction. And the authors also perform extra experiments to show that the proposed method is more robust to noisy data compared to the state-of-art.  I found no obvious flaw in this paper during review, good presentation and technical quality.